# Is there a bilingual advantage in auditory attention among children? A systematic review and meta-analysis of standardized auditory attention tests

Wenfu Bao[1,2]*, Claude Alain[3,4,5,6], Michael Thaut[2,5,6,7], Monika Molnar[1,2]

**1** Department of Speech-Language Pathology, University of Toronto, Toronto, Ontario, Canada, **2** Rehabilitation Sciences Institute, University of Toronto, Toronto, Ontario, Canada, **3** Rotman Research Institute, Baycrest Health Centre, Toronto, Ontario, Canada, **4** Department of Psychology, University of Toronto, Toronto, Ontario, Canada, **5** Institute of Medical Sciences, University of Toronto, Toronto, Ontario, Canada, **6** Music and Health Science Research Collaboratory, University of Toronto, Toronto, Ontario, Canada, **7** Faculty of Music, University of Toronto, Toronto, Ontario, Canada

* wenfu.bao@mail.utoronto.ca

**Data Availability Statement:** All relevant data are within the paper and its Supporting information files.

## Abstract

A wealth of research has investigated the associations between bilingualism and cognition, especially in regards to executive function. Some developmental studies reveal different cognitive profiles between monolinguals and bilinguals in visual or audio-visual attention tasks, which might stem from their attention allocation differences. Yet, whether such distinction exists in the auditory domain alone is unknown. In this study, we compared differences in auditory attention, measured by standardized tests, between monolingual and bilingual children. A comprehensive literature search was conducted in three electronic databases: OVID Medline, OVID PsycInfo, and EBSCO CINAHL. Twenty studies using standardized tests to assess auditory attention in monolingual and bilingual participants aged less than 18 years were identified. We assessed the quality of these studies using a scoring tool for evaluating primary research. For statistical analysis, we pooled the effect size in a random-effects meta-analytic model, where between-study heterogeneity was quantified using the $I^2$ statistic. No substantial publication bias was observed based on the funnel plot. Further, meta-regression modelling suggests that test measure (accuracy vs. response times) significantly affected the studies' effect sizes whereas other factors (e.g., participant age, stimulus type) did not. Specifically, studies reporting accuracy observed marginally greater accuracy in bilinguals ($g = 0.10$), whereas those reporting response times indicated faster latency in monolinguals ($g = -0.34$). There was little difference between monolingual and bilingual children's performance on standardized auditory attention tests. We also found that studies tend to include a wide variety of bilingual children but report limited language background information of the participants. This, unfortunately, limits the potential theoretical contributions of the reviewed studies. Recommendations to improve the quality of future research are discussed.

**Funding:** This work was funded by the Natural Sciences and Engineering Research Council of Canada (www.nserc-crsng.gc.ca; RGPIN-2019-06523) awarded to M.M. The funder had no role in study design, data collection and analysis, decision to publish, or preparation of the manuscript.

**Competing interests:** The authors have declared that no competing interests exist.

# Introduction

Over the past decade, the growing body of research investigating the effects of bilingualism on cognition suggests that bilingual experience can shape the brain and cognitive systems [1, 2]. Some studies [3, 4] demonstrate that bilingual participants outperform their monolingual counterparts on a wide range of cognitive tasks while controlling for other factors, which is often interpreted as a bilingual advantage. For example, bilingual children develop conflict resolution ability earlier than monolinguals and perform better on memory tasks based on executive control [3] (but see [5] for equal performance in inhibitory tasks).

Cognition has many facets, among which executive function (EF) is most frequently measured in bilingual developmental studies [6]. Recent meta-analyses present mixed results about whether bilingualism confers an EF benefit in children. For instance, Gunnerud et al. [7] analyzed 143 studies examining different EF components in children aged 18 months through 14.5 years; no bilingual advantage was found in overall EF after adjusting for publication bias. Similarly, Lowe et al. [8] reported a negligible effect of bilingualism on overall EF (*g* = -0.04) in children aged between 3 and 17 years. It is worth noting that when addressing attention, nearly all studies included in these meta-analyses have focused on the (audio-)visual domain, and very few measure auditory attention through tools like behavioral tasks or standardized tests. Using the Bayesian statistical approach, Grundy [9] revealed that when group differences do appear on EF tasks, bilinguals outperform monolinguals far more likely than chance. Given these discrepant findings, the bilingual advantage appears small at best and might be subject to specific circumstances [10]. However, what these "circumstances" are remains unclear.

Addressing whether a general bilingual advantage exists is beyond the scope of this study. We do not intend to engage in a dichotomous discussion to this debate, as it oversimplifies the effects of bilingualism on cognitive development. Nevertheless, we acknowledge that bilingualism can exert influence on cognition at least in some types of bilinguals, which has been supported by empirical studies [11, 12]. Further, attention has been hypothesized to be a plausible domain responsible for the bilingualism effects [2]. That is, the habitual use of two languages over years possibly enables bilinguals to become more practiced in managing conflicts and controlling attention. In particular, early bilingual exposure can affect how attention is allocated to the environment. For instance, infants growing up in bilingual homes pay more attention to subtle environmental differences [13, 14], which could improve their attentional processing [15].

However, a definition of attention remains vague in cognitive literature [16, 17]. Here, we conceptualize it as a system with three primary components: sustained attention, selective attention, and executive control [18, 19]. Specifically, based on Petersen and Posner [18], *sustained attention* (or alerting) refers to maintaining alertness over a long period of time, which is usually measured in the form of vigilance tasks that involve monitoring a target stimulus interspersed with non-targets. *Selective attention* (or orienting) refers to the ability to select certain input for enhanced processing while suppressing other irrelevant information, and thus is often measured by focusing on the target stimuli and ignoring the distractors. Lastly, *executive control* deals with resolving conflicts, shifting attention, and regulating thoughts and behavior. These components are also associated with different neural substrates in the human brain [18].

In bilingualism research, attention was not clearly conceptualized prior to a recent paper by Bialystok and Craik [16]. According to it, lifelong bilingual experience enhances attentional control, which is defined as a repertoire of processing operations that higher-level cognition utilizes to fulfill various goals [16]. However, visual or audio-visual attention has been

primarily studied in the past, whereas auditory attention has received little consideration. Though auditory attention is important to language processing and development: language is often processed through the auditory domain alone (e.g., speech perception), and infants start learning about language in utero without any visual support [20]. Likewise in the visual domain, bilingualism might also affect how auditory attention is allocated in bilingual children.

The current systematic review and meta-analysis assess whether there are reliable differences in auditory attention, measured by standardized tests, between monolingual and bilingual children. An initial search in our laboratory indicates that standardized tests are often used to assess children's auditory attention in research and clinical settings [21], and are an ideal comparison across contexts given consistent administration guidelines. These tests use different experimental paradigms to target different auditory attention components. For example, the Go/No-Go task is often employed to assess sustained attention, during which participants are asked to respond in some conditions but not to respond in others. Accordingly, depending on the task, different outcome measures are reported, such as response speed and accuracy. In addition, these tests use different types of auditory stimuli, which are either linguistic (e.g., syllables, words) or non-linguistic (e.g., tones, animal sounds; see Results for further information).

As attention might support monolingual and bilingual development differently [2], investigating auditory attention in monolingual and bilingual children illuminates different adaptations to their language environments. Additionally, examining the bilingualism effects on auditory attention addresses the current literature gap by focusing on the auditory domain, which has been overlooked in prior attention research. Considering evidence on bilingualism modulating audio-visual speech processing [22], our prediction is that if auditory attention development is shaped by bilingual experience, bilingual children might have more accurate and/or faster responses than their monolingual counterparts in standardized tests. We are also interested to explore whether the difference would vary by the attention components assessed.

An additional goal of our study is to determine whether certain bilingual characteristics (e.g., age of acquisition, language proficiency) mitigate the potential differences between monolinguals and bilinguals. Since the bilingualism effects are more evident among those with higher language proficiency and greater exposure [11, 12], we hypothesize that simultaneous bilinguals (i.e., children who learn both languages before the age of three) would more likely show enhanced auditory attention than sequential bilinguals (i.e., children who learn additional languages after the age of three). Finally, given that it is unclear whether bilingualism exerts influence beyond linguistic domain and standardized tests vary by stimulus type (linguistic vs. non-linguistic), we assess if different auditory stimuli affect attention performance in monolingual and bilingual children.

## Materials and methods

This study is part of a larger systematic review that investigates auditory attention development from infancy to adolescence, in which the population's language experience (i.e., monolingual vs. bilingual) and research methods are not controlled. The review protocol was registered a priori with OSF [21]. We followed the *Cochrane Handbook for Systematic Reviews of Interventions* [23] as the methods guidance, and developed the search strategy in consultation with a health sciences librarian. A PRISMA (Preferred Reporting Items for Systematic Reviews and Meta-Analyses) flow diagram illustrating the screening process is presented in Fig 1 (see more details in S1 Checklist). The methods described below are specific to our current (more focused) literature search embedded in Bao and Molnar [21].

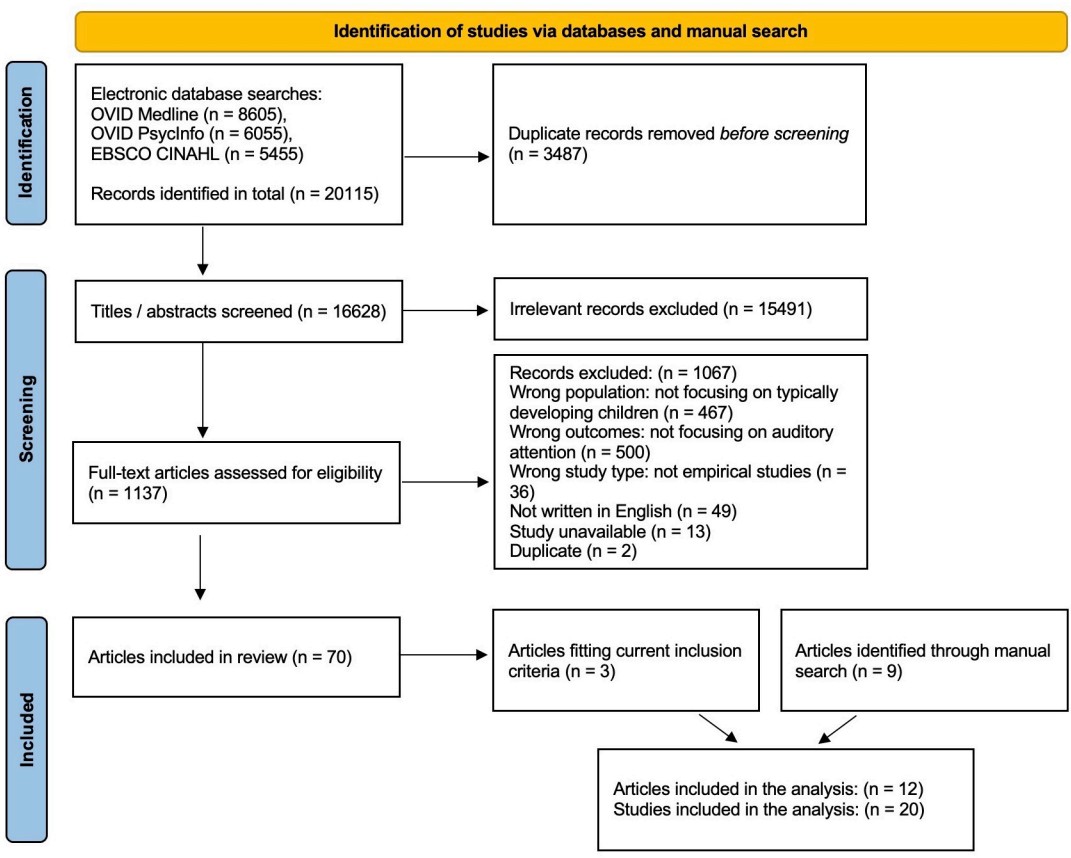

**Fig 1. PRISMA flow diagram presenting the study screening process.**

## Eligibility criteria

To include the most relevant studies on auditory attention measured by standardized tests, we adhered to the following criteria when determining a study's eligibility:

- Studies that used standardized tests to measure auditory attention were included. *Standardized tests* are norm-referenced tests administered and scored in a consistent manner. Studies using non-standardized measures or focusing on visual attention and other cognitive abilities were excluded.

- Studies that tested typically developing individuals below the age of 18 and had a monolingual group and a bilingual group were included. The bilingual group should be raised in bilingual families (i.e., at least one parent speaks another language than English to the child) or study in immersion schools where most of the curriculum is taught in an unfamiliar language. Studies only having monolingual or bilingual participants were excluded.

- Studies that controlled for participants' age and socio-economic status (SES) were included, as both factors can affect children's cognitive measures independently [24].

- Published, empirical studies that reported primary results in peer-reviewed articles were included.

- Only articles written in English were included.

## Information sources and search

A comprehensive literature search was conducted in three electronic databases: OVID Medline, OVID PsycInfo and EBSCO CINAHL. Search terms were defined after consulting the librarian and tailored to each database (see S1–S3 Tables). The search contained articles from the databases' start date up to March 8, 2023. Based on the search result of the larger systematic review, we identified all studies that compared monolingual and bilingual performance on standardized auditory attention tests. In addition, a manual search was performed by checking the reference lists of relevant articles, and an updated manual search was undertaken in Google Scholar to identify more studies.

## Study selection

Records and data were managed using the Covidence systematic review software [25]. Duplicates were identified and removed during reference importing and screening. Two independent reviewers screened studies in two phases: (1) title/abstract screening required to indicate "Yes," "No" or "Maybe" for relevance; (2) full-text screening required to indicate "Include" or "Exclude" for eligibility and specify the exclusion reasons. If discrepancies occurred, a third reviewer was called to resolve the conflicts.

## Data collection

For eligible studies, two independent reviewers extracted the following data items on Covidence: (1) sample size (monolingual vs. bilingual); (2) participant age; (3) language background (first and second language), bilingual type (simultaneous vs. sequential), and bilingualism assessment; (4) standardized tests used to assess auditory attention; (5) test measure (accuracy vs. response times or RTs); (6) stimulus type (linguistic vs. non-linguistic); (7) auditory attention components measured; (8) reporting of SES. In the case of unclear or missing items, we contacted the investigators to confirm and obtain additional information. Afterwards, a third reviewer compared the data extracted by the two reviewers and built consensus: for items where there was a conflict, a final decision was made by selecting or entering the most accurate response. Then the consensus data was exported for analysis.

## Quality assessment

The methodological quality of the studies was assessed by two independent reviewers using a modified version of the *Standard Quality Assessment Criteria for Evaluating Primary Research Papers*: *Quality Scoring for Quantitative Studies* (i.e., the "QualSyst" tool; [26]). Specifically, each study was evaluated according to 11 items based on description or reporting of objective, study design, participant selection, participant characteristics, outcome measure, analytic methods, estimate of variance, results, and conclusions, as well as sample size determination and confounding factors control. Each item was scored depending on the degree to which the specific criteria were met ("yes" = 2, "partial" = 1, "no" = 0), and then a summary score was calculated for each study and averaged between reviewers as the final rating. All studies received a score above 0.8, indicating high quality. Further, we used this tool to identify limitations of the reviewed studies and provide recommendations for future research.

## Data analysis

Apart from a narrative synthesis of included data, we performed a meta-analysis on the pooled effect size using the "meta" package (version 5.2–0) [27] in *R* [28]. Specifically, we extracted raw effect size data in the form of means and standard deviations of the two groups

(monolingual and bilingual) from the included studies. In view of insufficient details being reported, we contacted the original investigators to obtain them or used the WebPlotDigitizer tool [29] to extract data from the graphs (e.g., in Krizman et al. [30]). As we anticipated considerable between-study heterogeneity, a random-effects model was built using the "metacont" function to pool effect sizes. Given that our effect size data was continuous, we used the restricted maximum-likelihood estimator [31] to calculate the heterogeneity variance $\tau^2$, and the Hedges method to calculate the standardized mean difference (i.e., Hedges' $g$). To control for uncertainty in heterogeneity estimates, we used Knapp-Hartung adjustments [32] to calculate the confidence interval (CI) around the pooled effect. In terms of quantifying heterogeneity, we reported the $I^2$ statistic (i.e., percentage of variability not caused by sampling error) [33] along with its confidence intervals, as suggested in [34].

To further explore sources of statistical heterogeneity, we built mixed-effects meta-regression models using the "metafor" package (version 3.0–2) [35]. Specifically, the dependent variable was the unbiased estimate of the population effect size Hedges' $g$, calculated for each study. The independent variables were test measure (binary: accuracy vs. RTs), stimulus type (binary: linguistic vs. non-linguistic), participant age (continuous, in years) and attention components (categorical: selective attention, sustained attention, executive control, or auditory attention overall). To evaluate the effects of these predictors, we conducted model comparison in a forward stepwise manner: a likelihood ratio test was performed using the *anova* function to compare a reduced model and a full model, which had one additional component. Specifically, we inspected the estimated $p$-value and Akaike's Information Criterion (AIC) value to assess model performance (Bayesian Information Criterion or BIC is preferred over AIC when the heterogeneity is large in the studies [36]). The full model was favored only when the difference was significant as indicated by the $p$-value (less than the conventional threshold of 0.05) and when it provided a better fit for the data as suggested by lower AIC value. For categorical variables that significantly predicted the effect size, subgroup analyses were conducted using the "meta" package and forest plots were generated to visualize the effects. Of note, a limitation of meta-regression is that it describes an observational association across studies rather than a causal relationship; thus its findings should be interpreted with caution [37, 38].

Furthermore, publication bias was analyzed for the included studies by means of the funnel plot, created using the "meta" package. The funnel plot displays the studies' effect size against its standard error. Usually, symmetry indicates the absence of publication bias, which is reflected by data points scattered around the mean effect size forming an upside-down funnel [27]. This symmetry was further quantified through the Egger's regression test.

## Results

### Synthesis

The database search yielded 634 records for title/abstract screening. Forty articles remained for full-text review, and three articles qualified for extraction. Our manual search contributed another nine articles, four of which did not appear in the database search as they were not indexed. In total, 12 articles were extracted, which included 20 studies (see Table 1). Note that studies from the same article represent independent effect sizes, because multiple factors, such as age, outcome measure, stimulus type, etc. were considered across these studies.

**Population characteristics.** With an age range between 5 to 14 years, participants can be categorized into preschoolers (5–6 years), primary school-aged children (7–11 years), and young adolescents (12–14 years). This categorization has considered different education systems across countries. For example, while primary school starts at the age of four in the Netherlands, the first two years are comparable to kindergarten. Therefore, the five-year-old

**Table 1. Descriptive statistics of included studies.**

| Study | Country | Monolingual (n) | Bilingual (n) | Age (mean; years) | First language; second language | Bilingual type | Bilingualism assessment | Standardized test | Test measure | Stimulus type | Attention component | SES |
|---|---|---|---|---|---|---|---|---|---|---|---|---|
| Barbu et al. 2019 | Belgium | 57 | 59 | 6 | French; English | Sequential* (French children enrolled in an immersion program since the age of 5) | Customized language background questionnaire | KiTAP: The Owls | RT | Non-linguistic: animal sounds | Selective attention | Matched for two groups, from diverse levels |
| Boerma et al. 2017 | The Netherlands | 32 | 32 | 5 | Turkish, Tarifit-Berber, Arabic; Dutch | Sequential* (children learned Dutch as an L2 since the age of 4) | PaBiQ | IVA + Plus: Auditory task | Accuracy | Linguistic: numbers | Sustained attention | No group difference |
| Foy & Mann 2014 (1) | USA | 30 | 30 | 5 | Spanish; English | Simultaneous* (American children exposed to Spanish since at least 12 months of age) | Customized language background questionnaire; Language Dominance Survey from EOWPVT-SBE | ACPT-P (modified): Verbal Go/No-Go task | RT | Linguistic: speech syllables | Auditory attention (sustained attention, executive control) | Matched for two groups, all from low status |
| Foy & Mann 2014 (2) | | | | | | | | | | | | |
| Foy & Mann 2014 (3) | | | | | | | | ACPT-P (modified): Nonverbal Go/No-Go task | | Non-linguistic: animal and nature sounds | | |
| Foy & Mann 2014 (4) | | | | | | | | | | | | |
| Garratt & Kelly 2007 | UK | 27 | 27 | 7 | English; Bengali, Urdu/Punjabi, Malay, Arabic | Not reported | Customized language background questionnaire | NEPSY: Auditory Attention and Response Set | Accuracy | Linguistic: words | Auditory attention | Comparable between two groups, most from low status |
| Karlsson et al. 2015 (1) | Finland | 25 | 24 | 7 | Swedish; Finish and other ("other" second languages were not specified in the study) | Simultaneous (almost all bilingual children learned both languages by the age of 3) | Parental report | NEPSY-II: Auditory Attention | Accuracy | Linguistic: words | Selective and sustained attention | Matched for two groups, except that bilinguals had a higher level in the younger sample |
| Karlsson et al. 2015 (2) | | 23 | 27 | 11 | | | | | | | | |
| Karlsson et al. 2015 (3) | | 25 | 24 | 7 | | | | NEPSY-II: Response Set | | | Executive control | |
| Karlsson et al. 2015 (4) | | 23 | 27 | 11 | | | | | | | | |

*(Continued)*

**Table 1.** (Continued)

| Study | Country | Monolingual (n) | Bilingual (n) | Age (mean; years) | First language; second language | Bilingual type | Bilingualism assessment | Standardized test | Test measure | Stimulus type | Attention component | SES |
|---|---|---|---|---|---|---|---|---|---|---|---|---|
| Krizman et al. 2012 | USA | 25 | 23 | 14 | Spanish; English | Simultaneous (children had their first exposure to both languages about the age of 3) | LEAP-Q | IVA + Plus: Auditory task | Accuracy | Linguistic: numbers | Sustained and selective attention | Matched for two groups |
| Krizman et al. 2014 | USA | 27 | 27 | 14 | Spanish; English | Simultaneous (children learned both languages before/about the age of 3) | LEAP-Q; parental report | IVA + Plus: Auditory task | Accuracy | Linguistic: numbers | Attentional control | 44% of monolinguals and 59% of bilinguals from low status |
| Kwakkel et al. 2021 | The Netherlands | 80 | 89 | 5 | Dutch; English | Sequential* (Dutch children enrolled in a bilingual program since the age of 4) | Parental questionnaire | ACPT: Go/No-Go task | Accuracy | Non-linguistic: tones | Sustained attention | Matched for two groups, from high status overall |
| Nicolay & Poncelet 2013 | Belgium | 51 | 53 | 8 | French; English | Sequential* (French children enrolled in an immersion program since the age of 5) | Not explicitly assessed | KiTAP: The Owls | RT | Non-linguistic: animal sounds | Selective attention | Matched for two groups, from medium and high levels |
| Nicolay & Poncelet 2015 | Belgium | 50 | 51 | 8 | French; English | Sequential* (French children enrolled in an immersion program since the age of 5) | Not explicitly assessed | KiTAP: The Owls | RT | Non-linguistic: animal sounds | Selective attention | Matched for two groups, from diverse levels |
| Simonis et al. 2020 (1) | Belgium | 129 | 156 | 12 | French; Dutch, English | Sequential* (French children enrolled in an immersion program since the age of 5) | Background information questionnaire | TEA-Ch: Code Transmission (adapted) | Accuracy | Linguistic: numbers | Sustained attention | Bilinguals had higher SES |
| Simonis et al. 2020 (2) | | 153 | 173 | | | | | TAP: Divided Attention (adapted) | Accuracy | Non-linguistic: tones | Selective attention | |
| Simonis et al. 2020 (3) | | | | | | | | | RT | | | |

(Continued)

**Table 1.** (Continued)

| Study | Country | Monolingual (*n*) | Bilingual (*n*) | Age (mean; years) | First language; second language | Bilingual type | Bilingualism assessment | Standardized test | Test measure | Stimulus type | Attention component | SES |
|---|---|---|---|---|---|---|---|---|---|---|---|---|
| Strydom et al. 2022 | South Africa | 20 | 20 | 7 | Non-English (first languages were not specified in the study); English | Simultaneous* (children learned English as a second language before the age of 3) | Not explicitly assessed | SAAT: Words Intelligibility by Picture Identification in quiet | Accuracy | Linguistic: words | Selective attention | Same for two groups |

PaBiQ: Parents of Bilingual Children Questionnaire. EOWPVT-SBE: Expressive One Word Picture Vocabulary Test–Spanish-English Bilingual Edition. LEAP-Q: Language Experience and Proficiency Questionnaire. KiTAP: Test for Attentional Performance for Children; IVA + Plus: Integrated Visual and Auditory Continuous Performance Test; ACPT: Auditory Continuous Performance Test; ACPT-P: Auditory Continuous Performance Test-Preschoolers; TEA-Ch: Test of Everyday Attention for Children; TAP: Test for Attentional Performance; NEPSY(-II): A Developmental Neuropsychological Assessment (Second Edition); SAAT: Selective Auditory Attention Test.

Numbers following the names signify individual studies from the same article, which considers participant age, test measure, stimulus type, and attention components.

*Asterisk indicates that the bilingual type was not directly reported by the investigators but inferred from the original articles.

children in Boerma et al. [39] and Kwakkel et al. [40] were treated as preschoolers. Table 1 presents a detailed description of all participants. Taking language background for an example, there are a great variety of first and second languages, with French-English and Spanish-English being the most common language combinations within bilinguals.

However, quality assessment suggests that there was not sufficient information reported about bilingual characteristics in the reviewed papers. For instance, bilingual type (simultaneous vs. sequential) was rarely reported, except that Karlsson et al. [41] and Krizman et al. [30, 42] recruited "early bilinguals." Usually, bilingual children were from immigrant families learning the societal language as their second language (L2) [39, 43], or enrolled in immersion schools acquiring a foreign language [44–47]. Note that for longitudinal studies, we considered participants' language condition when they were tested. For example, Nicoley and Poncelet [46] collected data at two time points; data from the second one (i.e., three years after immersion school enrollment) were included only, because children cannot be deemed bilingual yet during the initial testing when they just started the program. Generally, immersion school children were considered sequential bilinguals, as most of them were from a monolingual environment and did not learn an L2 until they started school. Furthermore, bilingual experience was often evaluated using parental questionnaires that focus on one aspect (e.g., language use, exposure, or proficiency), which led to inconsistent definitions of bilinguals and the difficulty of comparing them across studies.

SES was reported in all studies, and mostly assessed through education level of the mother or of both parents (parental occupation was also used occasionally, e.g., Garratt and Kelly [43]). Monolingual and bilingual participants had a comparable SES across articles, except in Simonis et al. [47] where the bilingual group had a higher SES than the monolingual group. Given no significant difference in test performance between the two groups, this article was included in the final analysis.

**Test characteristics.** Various standardized tests (see Table 1) were used to assess auditory attention. Considering how they were described in the included studies, we categorized them into the following three groups: (1) Test for Attentional Performance (TAP) [48], Test of Attentional Performance for Children (KiTAP) [49], and Selective Auditory Attention Test (SAAT) [50]: assess *selective attention*; (2) Integrated Visual and Auditory Continuous Performance Test (IVA + Plus) [51], Auditory Continuous Performance Test for Preschoolers (ACPT-P) [52], and Test of Everyday Attention for Children (TEA-Ch) [53]: assess *sustained attention*; (3) Developmental Neuropsychological Assessment (NEPSY) [54] and its second version (NEPSY-II) [55]: assess *auditory attention overall* (selective and sustained attention, and executive control). For tests evaluating audio-visual attention, we included data from subtests using only auditory stimuli. Further, we found that both linguistic and non-linguistic stimuli were utilized. For example, some tests used non-linguistic stimuli like auditory tones (TAP) and animal sounds (KiTAP; e.g., [44, 47]), whereas others employed linguistic stimuli such as numbers (IVA + Plus; TEA-Ch), speech syllables (ACPT-P), and words (NEPSY, NEPSY-II, SAAT; e.g., [30, 39, 42, 56]). Though it remains unknown whether different types of auditory stimuli affect attentional performance.

Moreover, some researchers used translated versions (e.g., Swedish version of NEPSY-II in [41]) or adapted the tests [47], and reported the assessed attention components differently. For example, modelled after ACPT-P, Foy and Mann [57] used an auditory Go/No-Go task with two blocks. We infer that the first block assessed sustained attention and the second one evaluated executive control, as the authors did not explicitly report the attention components. Using the same task, however, Kwakkel et al. [40] stated that they measured "sustained attention."

Another source of variance we observed involved the test measure reported. All studies presented either accuracy or RTs, except that Simonis et al. [47] did both. Nevertheless, how accuracy was reported in different studies was not always consistent, even for the same test. For instance, Garratt and Kelly [43] reported standard scores for the Auditory Attention and Response Set subtests of NEPSY together, whereas Karlsson et al. [41] reported raw scores for each subtest of NEPSY-II individually. Simonis et al. [47] used decimal numbers to indicate accuracy (i.e., 1 = 100% accuracy), whereas Kwakkel et al. [40] used reversed omission scores. Despite these discrepancies, for all accuracy statistics in this meta-analysis, larger values indicate better auditory attention. However, we did not include data that were presented in other forms and could not be converted to accuracy scores or proportions, e.g., misses in Foy and Mann [57] and errors in Barbu et al. [44].

## Meta-analysis

Effect size was pooled in the meta-analysis. According to the random-effects model (see S4 Table), the between-study heterogeneity variance was estimated at $\tau^2$ = 0.09 (95% CI: 0.03–0.29), with an $I^2$ value of 65.8% (95% CI: 45.1–78.6%) that indicates moderate to substantial heterogeneity. Meta-regression modelling was further implemented to address the heterogeneity effect between studies. We first built a mixed-effects model, with each study's effect size as the dependent variable and test measure as the moderator. Model summary suggested that test measure significantly influenced the studies' effect size ($p$ = 0.0067; see S5 Table). Further, results of a subgroup analysis confirmed that there was a significant difference in effect size between studies reporting accuracy (i.e., accuracy studies) and studies reporting RTs (i.e., RT studies; $p$ = 0.0014; see S6 Table). Fig 2 shows a forest plot stratified by test measure: bilingual children presented more accurate responses in accuracy studies ($g$ = 0.10), but slower latency in RT studies ($g$ = -0.34) than their monolingual counterparts. However, results favored monolingual children ($g$ = -0.09) in standardized auditory attention tests when taking both measures together.

To investigate the effects of other variables, we divided our data into two subsets (i.e., accuracy studies and RT studies) and analyzed them separately. For accuracy studies ($n$ = 12), a meta-regression model with participant age as the predictor was built. Age did not influence the effect size significantly ($p$ = 0.1282; see S7 Table). However, a positive trend favoring young bilingual adolescents was observed (see S1 Fig). Following the same approach, we tested the effects of stimulus type and attention components, but neither of them was significant (stimulus type: $p$ = 0.9550; attention components: $p$ = 0.4865; see S8 and S9 Tables). For RT studies ($n$ = 8), none of the variables above was significant (participant age: $p$ = 0.2056, stimulus type: $p$ = 0.5494, attention components: $p$ = 0.9396; see S10–S12 Tables). In addition, there was not substantial publication bias in our data according to the funnel plot (S2 Fig), which was further confirmed by the result of Egger's regression test ($p$ = 0.414; see S13 Table). However, we did not include unpublished research, which is a potential limitation.

## Discussion

The current systematic review and meta-analysis analyzed 20 studies that compared monolingual and bilingual children's performance on standardized auditory attention tests. Results suggest that test measure was significantly related to differences in effect sizes: accuracy studies ($n$ = 12) indicated marginally greater accuracy in bilinguals, whereas RT studies ($n$ = 8) indicated faster responses in monolinguals. However, no other factors (i.e., participant age, stimulus type, attention components) resulted in significant differences or interactions. Overall, there was little difference between monolingual and bilingual children's performance on

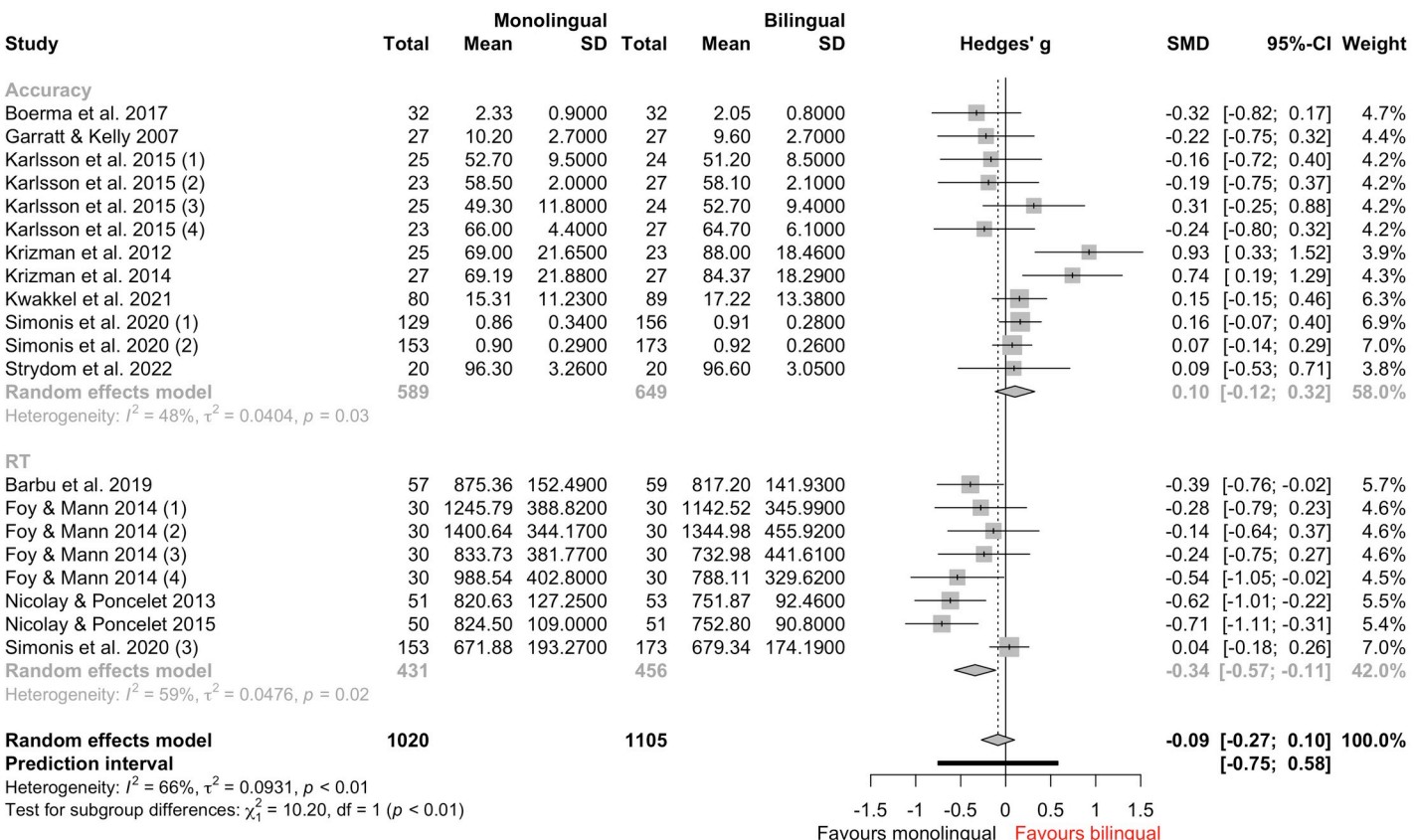

**Fig 2. Forest plot stratified by test measure, based on random-effects meta-analytic model analysis.** The upper and lower panels display the results for accuracy studies ($n = 12$) and RT studies ($n = 8$), respectively.

standardized auditory attention tests. This finding is consistent with those of Gunnerud et al. [7] and Lowe et al. [8], though both of them analyzed tasks focusing on visual attention.

In answer to the question in the title, a bilingual advantage in auditory attention, when measured using standardized tests, likely does not exist or is limited to certain conditions (e.g., accuracy measure or more proficient bilinguals). To be fair in our interpretation of this finding, two points should be considered: (1) a very heterogeneous bilingual population was synthesized across studies, wherein bilingualism was poorly assessed; (2) most standardized tests are developed for monolingual English speakers, thus they might not be suitable to measure bilinguals. Importantly, our result fits well within the theoretical framework discussed in Paap [58], who highlights that the bilingual advantage either does not exist or is restricted to very specific circumstances. Our work specifically suggests that a bilingual advantage is not observable when measured through standardized auditory attention tests.

In the reviewed papers, standardized auditory attention tests were administered to participants as young as five years of age, and to children (5–11 years) more often than to young adolescents (12–14 years). Monolingual children seemingly had faster RTs, but bilingual children showed somewhat higher accuracy. Prior work from other domains than auditory attention has presented mixed findings about the RT difference between groups. For instance, Bialystok et al. [59] used the Simon task—a non-linguistic interference task—and observed faster latency in bilingual children, although this effect was later only found when the demands for

inhibitory control were high [60]. However, studies using language processing tasks show longer RTs in bilingual adults. For example, bilinguals have slower RTs to target words during a lexical decision task [61], and perform more slowly in a picture naming task than monolinguals [62]. These linguistic tasks specifically measure cross-language interference and switching, and bilinguals' slower RTs have been often associated with their need to navigate more than one language system.

In our meta-analysis, bilingual children seemed to have slightly more accurate responses, but at the cost of longer latency relative to their monolingual peers. However, these findings should be interpreted with caution, because most accuracy and RT measures were derived from different studies across the reviewed articles. Only Simonis et al. [47] reported both measures, but neither accuracy nor latency showed group differences between monolingual and bilingual children in their study. To draw a firm conclusion on the relationship between accuracy and RTs, one should only consider studies that report both measures within the same group.

With regards to the measured auditory attention components, selective attention and sustained attention are mostly assessed, followed by executive control. While one may argue that it is hard to disentangle these components and measure them independently, we distinguished them by considering the specific tasks used by investigators and the attention model developed by Posner and colleagues [18]. We also noticed inconsistency across publications when reporting the measured components with a given test. For example, using the auditory section of IVA + Plus, Krizman et al. [42] measured "attentional control," whereas Boerma et al. [39] assessed "sustained attention." This observation is in line with Williams et al. [6] who also found variation in targeted cognitive abilities for the same task in bilingualism research.

Moreover, rarely were sufficient details about participants' bilingual background reported. This is problematic because differences in language experience (e.g., age of acquisition, proficiency; [4, 11, 63]) can affect cognitive performance. For example, more proficient bilinguals tend to exhibit a positive influence of bilingualism. Consistent with this, we observed a trend of the bilingual advantage toward adolescents relative to younger children, which can be explained by the fact that bilingual children become more proficient in their languages when growing older.

In addition, bilingualism was poorly assessed and hardly deemed a continuum. We identified two common themes: (1) most language background questionnaires fail to assess bilingualism comprehensively (i.e., including the use of, exposure to, and proficiency of each language; see [30]); (2) bilingual participants are usually recruited as a comparison group to monolinguals, without explaining inclusion criteria. This practice overlooks the nuances within bilinguals. For example, an English L2 learner from Spain is not comparable to a Spanish heritage speaker from the U.S., although both considered as Spanish-English bilinguals. Thus, variance in different bilingual communities should be considered, captured, and reported in future work.

In future studies, bilingualism should be considered as a continuum, and bilingual variables should be appropriately assessed and reported. This will facilitate our understanding of how various facets of bilingualism (e.g., age of acquisition, language proficiency, language exposure, language use, etc.) interact with auditory attention development. Moreover, further investigation should test monolinguals and bilinguals across age groups and report both accuracy and RTs. Given that most tests are designed for monolingual English speakers, we also advocate for developing tests that are normed to assess both monolinguals and bilinguals. Lastly, there was little consistency regarding the targeted attention components even for the same test across publications, which may be due to the lack of a clear definition of attention in the field itself.

Therefore, we need to consider further what attention along with its components refers to in (bilingualism) research and theoretical models.

## Supporting information

**S1 Fig. Forest plot stratified by participant age groups, based on random-effects model analysis.** Green lines represent accuracy studies, and blue lines represent RT studies.
(TIF)

**S2 Fig. Contour-enhanced funnel plot.** Effect size (i.e., standardized mean difference or SMD) is plotted against its standard error. Each dot represents an individual study. Grey-shaded areas indicate different *p*-value intervals.
(TIF)

**S1 Table. Search terms used in the electronic databases under the concept of "auditory attention".**
(DOCX)

**S2 Table. Search terms used in the electronic databases under the concept of "population".**
(DOCX)

**S3 Table. Search terms used in the electronic databases under the concept of "study types and methods".**
(DOCX)

**S4 Table. Random-effects meta-analytic model summary.**
(DOCX)

**S5 Table. Mixed-effects meta-regression model summary, with test measure as the moderator.**
(DOCX)

**S6 Table. Subgroup analysis result, stratified by test measure.**
(DOCX)

**S7 Table. Mixed-effects meta-regression model summary for accuracy studies, with participant age as the moderator.**
(DOCX)

**S8 Table. Mixed-effects meta-regression model summary for accuracy studies, with stimulus type as the moderator.**
(DOCX)

**S9 Table. Mixed-effects meta-regression model summary for accuracy studies, with attention components as the moderator.**
(DOCX)

**S10 Table. Mixed-effects meta-regression model summary for RT studies, with participant age as the moderator.**
(DOCX)

**S11 Table. Mixed-effects meta-regression model summary for RT studies, with stimulus type as the moderator.**
(DOCX)

**S12 Table. Mixed-effects meta-regression model summary for RT studies, with attention components as the moderator.**
(DOCX)

**S13 Table. Egger's regression test result.**
(DOCX)

**S1 Checklist. PRISMA 2020 checklist.**
(DOCX)

## Author Contributions

**Conceptualization:** Wenfu Bao, Monika Molnar.

**Data curation:** Wenfu Bao.

**Formal analysis:** Wenfu Bao.

**Funding acquisition:** Monika Molnar.

**Investigation:** Wenfu Bao.

**Methodology:** Wenfu Bao.

**Supervision:** Monika Molnar.

**Writing – original draft:** Wenfu Bao.

**Writing – review & editing:** Wenfu Bao, Claude Alain, Michael Thaut, Monika Molnar.

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
