## [Decision Letter · Decision Letter 0]

20 Jun 2023

PONE-D-23-08778Is there a bilingual advantage in auditory attention among children? A systematic review and meta-analysis of standardized auditory attention testsPLOS ONE

Dear Dr. Bao,

Thank you for submitting your manuscript to PLOS ONE. After careful consideration, we feel that it has merit but does not fully meet PLOS ONE’s publication criteria as it currently stands. Therefore, we invite you to submit a revised version of the manuscript that addresses the points raised during the review process.

The manuscript has been evaluated by two reviewers, and their comments are available below.The reviewers have made a number of requests for clarification and additional information.Could you please carefully revise the manuscript to address all comments raised?

We look forward to receiving your revised manuscript.

Kind regards,

Steve Zimmerman, PhD

Associate Editor, PLOS ONE

Reviewers' comments:

Reviewer's Responses to Questions

**Comments to the Author**

1. Is the manuscript technically sound, and do the data support the conclusions?

Reviewer #1: Yes

Reviewer #2: Yes

2. Has the statistical analysis been performed appropriately and rigorously? 

Reviewer #1: No

Reviewer #2: I Don't Know

3. Have the authors made all data underlying the findings in their manuscript fully available?

Reviewer #1: Yes

Reviewer #2: Yes

4. Is the manuscript presented in an intelligible fashion and written in standard English?

Reviewer #1: Yes

Reviewer #2: Yes

5. Review Comments to the Author

Reviewer #1: Overall, I think this is an interesting systematic review and meta-analysis of the subject. However, given the small number of papers included in the final analysis, I am not sure all the detailed statistical tests are possible with the sample split in half (only 12 resp. 8 data points analysed with many predictors - why is this division into RT and accuracy studies necessary if you use effect sizes as a dependent variable?). In any case, the non-significance of the results cannot really be interpreted, so the conclusions (that there is no difference) should be tuned down.

Minor comments: The introduction could center a bit more on auditory attention and explain the different paradigms and stimuli used in this domain, so that the partitions used later in the analysis are understandable (p.ex. stimulus type, RT vs accuracy...). By the way, this type of catgorisation of studies should be better described and justified (non-linguistic stimuli for instance seems quite a broad category, why no further breakdown?). Furthermore, the discussion of hypotheses why there might be a bilingual advantage (or not) could be developed (p.ex. audio-visual effects in early speech perception....). The dichotomy between simultaneous and sequential bilinguals is not as clear as presented, current amount of exposure/ input seems to play a bigger role and should be at least mentioned (for immersion vs. family contexts, for instance, this would be radically different). In the data analysis part, it is not clear how the third reviwer "built consensus", and the model selection by anova vs. AIC/ BIC seems redundant (decide on one criterion, or explain better why they are all needed). Finally, it is not clear whether the bilingual/ monolingual advantages found are corrected for other factors (age/ SES), or whether that was done in the original studies.

Reviewer #2: PONE-D-23-08778

General comments: The ms presents the results of a meta-analysis of studies of auditory attention among monolingual and bilingual children. Overall, the ms reads well and presents interesting and informative findings. The following questions and comments should be addressed in a revision.

1. While the authors attempt to distance their study from the larger controversy concerning the bilingual advantage in higher-order cognition (which is probably wise), there are a few places in the ms where they could do better at avoiding unwelcome controversy. First, the authors use evaluative terminology to frame the literature in the Introduction and to describe findings in the Abstract and Discussion. Lines 36-38 in the Abstract, for example, the authors characterize differences between monolingual and bilinguals as relative benefits or advantages. Similarly, evaluative terminology appears in different places in the Introduction (lines 65-75; line 107; etc.), including a prediction on line 107 that bilinguals might perform “better” than monolinguals. What is “better” attention? If the authors do really want to steer clear of the controversy and focus on the science, eliminate evaluative terminology from the discussion of group differences. The authors also (somewhat duplicitously) claim they have no interest in engaging in the debate, but then appeal to Carlson and Meltzoff’s study on line 69 as some sort of incontrovertible evidence of the bilingual advantage. Please look at the C & M findings more closely. They compared 12 (!) bilinguals and 17 (!) monolinguals on 9 measures of EF and found no differences between the groups on any of the 9 measures. Only after correcting for differences in L1 – a highly suspect procedure given known L1 differences between monolingual and bilinguals -- did they find differences on 3 or the 9 EF tasks. As discussed in the literature (see Morton, 2015), this is hardly strong evidence of a group difference, let alone an advantage. Requires some rethinking.

2. While the authors should be commended for keeping the Introduction succinct, the motivation for the current analysis and in particular, the predictions, is a bit thin. Line 91, we read that “bilingualism might also affect how auditory attention is allocated in bilingual children.” Beyond the redundant reference to bilingualism in bilingual children, the reader is left wondering why? Is there some theory that would lead to this prediction? How do the predictions and goals described on Page 5 and 6 relate to the theoretical characterization of attention on Page 4 (lines 76-85)? There is no obvious connection.

3. From the chart in Figure 2, it appears that there are multiple dependencies between effect sizes included in the analysis including different effects from the same study and different effects from different studies from the same lab. The authors need to provide more information about the mixed-effects model they used in the analysis (see page 15) so the readership can evaluative whether these dependencies were properly modelled.

4. The authors need to clarify how measuring bilingualism as a continuum will reveal associations with measures of attention that are obscured when bilingualism is defined dichotomously (see line 364). Bear in mind, many studies comparing monolinguals and bilinguals are comparisons of extreme groups, with monolinguals having very little dual language experience and bilinguals having a whole lot. If there are no differences between extreme groups, what is examining the continuum of bilingualism going to reveal? Is the relation between bilingual language experience and attention non-linear, with the largest effects seen for groups with moderate levels of experience and smaller effects for the extreme groups? Also, what even is a continuum of bilingualism? As the authors so eloquently describe on lines 355-363, an English L2 learner from Spain is very different than a Spanish heritage speaker in the US. Is there really a single continuum that captures this variability? Requires further thought.

6. PLOS authors have the option to publish the peer review history of their article (what does this mean?). If published, this will include your full peer review and any attached files.

Reviewer #1: No

Reviewer #2: No

---

## [Author Response · Author response to Decision Letter 0]

1 Aug 2023

We would like to thank both Reviewers for their constructive comments and insightful feedback. Below is our point-by-point response to each comment. 

Reviewer #1 

Comment 1.1: 

Overall, I think this is an interesting systematic review and meta-analysis of the subject. However, given the small number of papers included in the final analysis, I am not sure all the detailed statistical tests are possible with the sample split in half (only 12 resp. 8 data points analysed with many predictors - why is this division into RT and accuracy studies necessary if you use effect sizes as a dependent variable?). In any case, the non-significance of the results cannot really be interpreted, so the conclusions (that there is no difference) should be tuned down. 

Response 1.1: 

We really appreciate these important observations. First, regarding the concern about the number of studies included in our meta-analysis, we believe that our sample size (n = 20) is sufficient in terms of statistical power. It is not uncommon for published meta-analyses to include a similar number of studies or even less (e.g., Greene, 1997; Huang, 2010; Paulavicius et al., 2020). Also, as a general rule of thumb, subgroup analysis can be conducted when the meta-analysis contains at least 10 studies (Harrer et al., 2021; Schwarzer et al., 2015). 

Second, we think that it would be conceptually unwise to include accuracy and RT in the same analysis, even though they are both represented by effect size values. Accuracy and RT correspond to two different aspects of stimuli processing in these auditory attentional tasks: decision signaled by a behavioral response vs. the amount of time it takes to reach the decision. The relationship between the dependent variables associated with these processes are not straightforward; moreover, different patterns of accuracy and RT values have been observed across monolinguals and bilinguals—which is relevant to our study (Morales et al, 2013; Sullivan et al., 2018). Our line of thinking is supported by the statistical analyses reported in the manuscript and highlighted below. 

As outlined in the “Results - Meta-analysis” section on page 15, test measure (i.e., accuracy vs. RT) significantly influenced effect size, when all studies (n = 20) were considered in constructing mixed-effects meta-regression models. Note that this significance level was much lower than the conventional threshold of 0.05, with a very small p-value of 0.0067 (see S4 Table). Furthermore, the fact that test measure was a significant effect moderator was also evidenced by the results of a subgroup analysis: there was a significant difference in effect size between accuracy studies and RT studies (p = 0.0014; see S5 Table), with distinct patterns being illustrated in the forest plot in Fig 2. 

To summarize, given the between-study heterogeneity significantly affected by test measure, we did not add more variables into the meta-regression model. Otherwise, the pooled effect size won’t be interpreted in a meaningful way. Rather, we split the data into accuracy studies and RT studies and analyzed the effects of other factors in each dataset separately. 

Finally, with regards to our conclusion, we had proceeded with caution and wrote that “a bilingual advantage in auditory attention, when measured using standardized tests, likely does not exist or is restricted to certain conditions (e.g., accuracy measure or more proficient bilinguals)” on page 17. To clarify, this conclusion is based on the observed effect size (e.g., g = 0.10 among accuracy studies; a positive trend favoring bilingual adolescents), rather than on any p-values associated with meta-regression modelling. Further, we also provided two considerations when interpreting our results on the same page: “(1) a very heterogeneous bilingual population was synthesized across studies, wherein bilingualism was poorly assessed; (2) most standardized tests are developed for monolingual English speakers, thus they might not be suitable to measure bilinguals.” Thus, there is probably little room left to further tune down our conclusion. 

References: 

Greene, J. P. (1997). A meta-analysis of the Rossell and Baker review of bilingual education research. Bilingual Research Journal, 21(2-3), 103–122. https://doi.org/10.1080/15235882.1997.10668656

Harrer, M., Cuijpers, P., Furukawa, T. A., & Ebert, D. D. (2021). Doing Meta-Analysis with R: A Hands-On Guide. Boca Raton, FL and London: Chapmann & Hall/CRC Press. 

Huang, S.-F. (2010). Effects of tasks and glosses on L2 incidental vocabulary learning: Meta-analyses. [Doctoral dissertation, Texas A&M University]. ProQuest Dissertations and Theses Global. 

Morales, J., Gómez-Ariza, C. J., & Bajo, M. T. (2013). Dual mechanisms of cognitive control in bilinguals and monolinguals. Journal of Cognitive Psychology, 25(5), 531–546. https://doi.org/10.1080/20445911.2013.807812

Paulavicius, A. M., Mizzaci, C. C., Tavares, D. R. B., Rocha, A. P., Civile, V. T., Schultz, R. R., Pinto, A. C. P. N., & Trevisani, V. F. M. (2020). Bilingualism for delaying the onset of Alzheimer’s disease: A systematic review and meta-analysis. European Geriatric Medicine, 11, 651–658. https://doi.org/10.1007/s41999-020-00326-x

Schwarzer, G., Carpenter, J. R., & Rücker, G. (2015). Meta-Analysis with R. London: Springer. 

Sullivan, M. D., Poarch, G. J., & Bialystok, E. (2018). Why is lexical retrieval slower for bilinguals? Evidence from picture naming. Bilingualism: Language and Cognition, 21(3), 479–488. https://doi.org/10.1017/S1366728917000694

Minor comments: 

Comment 1.2: 

The introduction could center a bit more on auditory attention and explain the different paradigms and stimuli used in this domain, so that the partitions used later in the analysis are understandable (p.ex. stimulus type, RT vs accuracy...). By the way, this type of catgorisation of studies should be better described and justified (non-linguistic stimuli for instance seems quite a broad category, why no further breakdown?). 

Response 1.2: 

Thank you for the feedback. To clarify, instead of addressing auditory attention up front, we approached it step by step in the Introduction. Specifically, we first ushered readers into bilingualism studies on cognition, then narrowed the focus to attention, and finally zeroed in on attention in the auditory domain. Although it may not seem centered on auditory attention at first glance, this “peel-the-onion” way of storytelling serves the purpose of introducing our research better. 

Following your suggestion, we have elaborated on the experiment paradigms, outcome measures, and stimuli when mentioning the standardized tests. The revised text reads as follows: 

“An initial search in our laboratory indicates that standardized tests are often used to assess children’s auditory attention in research and clinical settings [...]. These tests use different experimental paradigms to target different auditory attention components. For example, the Go/No-Go task is often employed to assess sustained attention, during which participants are asked to respond in some conditions but not to respond in others. Accordingly, depending on the task, different outcome measures are reported, such as response speed and accuracy. In addition, these tests use different types of auditory stimuli, which are either linguistic (e.g., syllables, words) or non-linguistic (e.g., tones, animal sounds; see Results for further information).” 

As to the categorization of linguistic vs. non-linguistic stimuli, we did not further break them down. This is because as we explained on page 6, we were interested in whether different stimulus type would affect attentional performance in monolingual and bilingual children. We asked this question given that previous studies have reported mixed results on the bilingualism effects in the non-linguistic domain. Moreover, there is little theoretical motivation that could guide us in terms of how to further break down non-linguistic stimuli. Thus, it is beyond our scope to investigate, for example, whether listening to animal sounds or auditory tones would affect participants’ performance in standardized tests between monolingual and bilingual children. 

Comment 1.3: 

Furthermore, the discussion of hypotheses why there might be a bilingual advantage (or not) could be developed (p.ex. audio-visual effects in early speech perception....). The dichotomy between simultaneous and sequential bilinguals is not as clear as presented, current amount of exposure/ input seems to play a bigger role and should be at least mentioned (for immersion vs. family contexts, for instance, this would be radically different). 

Response 1.3: 

Thank you very much for pointing these out. To support our hypothesis about the monolingual vs. bilingual children difference, we have referred to Pons et al. (2015) who demonstrated that bilingual experience modulates early audio-visual speech processing. The revised text reads as follows: “Considering evidence on bilingualism modulating audiovisual speech processing [19], our prediction is that if auditory attention development is shaped by bilingual experience, bilingual children might have more accurate and/or faster responses than their monolingual counterparts in standardized tests.” 

To acknowledge the effect of language exposure, we have revised the reasoning of our hypothesis as: “Since the bilingualism effects are more evident among those with higher language proficiency and greater exposure [11, 12], we hypothesize that simultaneous bilinguals (i.e., children who learn both languages before the age of three) would more likely show enhanced auditory attention than sequential bilinguals (i.e., children who learn additional languages after the age of three).” 

Comment 1.4: 

In the data analysis part, it is not clear how the third reviwer "built consensus", and the model selection by anova vs. AIC/ BIC seems redundant (decide on one criterion, or explain better why they are all needed). 

Response 1.4: 

Thank you for flagging this to us. To improve clarity, we have revised the sentence as “Afterwards, a third reviewer compared the data extracted by the two reviewers and built consensus: for items where there was a conflict, a final decision was made by selecting or entering the most accurate response. Then the consensus data was exported for analysis.” 

To assess model performance, we used the anova function of the “metafor” package to determine if one model has a better fit than the other. This function performs a likelihood ratio test; apart from the resulting p-value of the test, it provides statistics such as the AIC and BIC values associated with each model, which should also be considered in model comparison (Harrer et al., 2021). In particular, the AIC value, corrected for small samples, penalizes complex models with more predictors to avoid overfitting (Harrer et al., 2021). As we explained in the manuscript, “BIC is preferred over AIC when the heterogeneity is large in the studies [27]”; for both statistics, lower values indicate better model performance. Therefore, it is important to consult multiple statistics rather than solely relying on the p-value (see Haysey, 2019). 

To make it clearer that we used both criteria in model selection, we have rewritten the relevant text as: “Specifically, we inspected the estimated p-value and Akaike’s Information Criterion (AIC) value to assess model performance (Bayesian Information Criterion or BIC is preferred over AIC when the heterogeneity is large in the studies [27]). The full model was favored only when the difference was significant as indicated by the p-value (less than the conventional threshold of 0.05) and when it provided a better fit for the data as suggested by lower AIC value.” 

References: 

Halsey, L. G. (2019). The reign of the p-value is over: what alternative analyses could we employ to fill the power vacuum? Biology Letters, 15(5). http://doi.org/10.1098/rsbl.2019.0174

Comment 1.5: 

Finally, it is not clear whether the bilingual/ monolingual advantages found are corrected for other factors (age/ SES), or whether that was done in the original studies. 

Response 1.5: 

As we outlined in the eligibility criteria on page 7, “studies that controlled for participants’ age and socio-economic status (SES) were included.” More specifically, age was controlled in all of the original studies, as monolingual and bilingual children were compared within the same age group (see Table 1). In our meta-analysis, to explore the age effects, we included it as a variable of interest in statistical models and found that it did not significantly influence the effect size in accuracy studies and RT studies (see detailed results on page 16). 

In addition, we had explicitly stated that monolingual and bilingual children had a comparable SES across all original studies (see Table 1), “except in Simonis et al. [38] where the bilingual group had a higher SES than the monolingual group. Given no significant difference in test performance between the two groups, this article was included in the final analysis” (page 13). 

Reviewer #2 

General comments: The ms presents the results of a meta-analysis of studies of auditory attention among monolingual and bilingual children. Overall, the ms reads well and presents interesting and informative findings. The following questions and comments should be addressed in a revision. 

Response: Thank you very much for the feedback. We have addressed your questions and comments below and revised the manuscript accordingly. 

Comment 2.1: 

While the authors attempt to distance their study from the larger controversy concerning the bilingual advantage in higher-order cognition (which is probably wise), there are a few places in the ms where they could do better at avoiding unwelcome controversy. First, the authors use evaluative terminology to frame the literature in the Introduction and to describe findings in the Abstract and Discussion. Lines 36-38 in the Abstract, for example, the authors characterize differences between monolingual and bilinguals as relative benefits or advantages. Similarly, evaluative terminology appears in different places in the Introduction (lines 65-75; line 107; etc.), including a prediction on line 107 that bilinguals might perform “better” than monolinguals. What is “better” attention? If the authors do really want to steer clear of the controversy and focus on the science, eliminate evaluative terminology from the discussion of group differences. The authors also (somewhat duplicitously) claim they have no interest in engaging in the debate, but then appeal to Carlson and Meltzoff’s study on line 69 as some sort of incontrovertible evidence of the bilingual advantage. Please look at the C & M findings more closely. They compared 12 (!) bilinguals and 17 (!) monolinguals on 9 measures of EF and found no differences between the groups on any of the 9 measures. Only after correcting for differences in L1 – a highly suspect procedure given known L1 differences between monolingual and bilinguals -- did they find differences on 3 or the 9 EF tasks. As discussed in the literature (see Morton, 2015), this is hardly strong evidence of a group difference, let alone an advantage. Requires some rethinking. 

Response 2.1: 

We appreciate the points raised by the Reviewer. Indeed, the type of language we use to describe our thoughts is relevant. To minimize evaluative terminology, we have reworded text when addressing group differences, including the Abstract, Introduction, Results, and Discussion sections. Nevertheless, it should be also noted that sometimes it is impossible to avoid evaluative terminology when describing the different outcomes produced by monolinguals and bilinguals. In the tests reviewed here, more accurate and/or faster outcomes are interpreted as “better” performance. For example, in the Abstract, we have revised our findings as: “Specifically, studies reporting accuracy observed marginally greater accuracy in bilinguals (g = 0.10), but those reporting response times indicated faster latency in monolinguals (g = -0.34).” Similarly, we have changed multiple places in

---

## [Decision Letter · Decision Letter 1]

24 Oct 2023

PONE-D-23-08778R1Is there a bilingual advantage in auditory attention among children? A systematic review and meta-analysis of standardized auditory attention testsPLOS ONE

Dear Dr. Bao, 

Thank you for submitting your manuscript to PLOS ONE. After careful consideration, we feel that it has merit but does not fully meet PLOS ONE’s publication criteria as it currently stands. Therefore, we invite you to submit a revised version of the manuscript that addresses the points raised during the review process.  Unfortunately both Reviewer1 and myself feel that this revised version requires more revision  than than the original manuscript.  In striving to meet the requirements of revision  you  did not acknowledge recent meta analysis findings on how bilingualism is not linked to executive function. Your paper suggests that executive function is a mechanism  by stating research suggest bilinguals are advantaged in “conflict resolution” and executive control tasks relative to monolinguals. Further you need to think about your claim that attention is not clearly defined with respect to bilingualism. You need to either revise this claim or make an evidence-based argument for why this is the case. Think carefully about your discussion . The results of your systematic review suggest no bi-lingual advantage for auditory attention..You need to provide a definition of heterogeneity in the methods.Reviewer 2 raises a number of points about your results section. There needs to be a restructuring of the statistical analysis section by commencing with the meta analysis, the heterogeneity etc and finish with the meta-regression. Yiu must also report confidence intervals. These are some of the key points but you must either address all points raised by the reviewers or provide a good reason why you are not addressing them.  Addressing these points will greatlyimprove the quality of your systematic analysis. Please ensure that your decision is justified on PLOS ONE’s publication criteria and not, for example, on novelty or perceived impact.

Please submit your revised manuscript by Dec 08 2023 11:59PM. You can also submit before this deadline if you wish. If you will need more time than this to complete your revisions, please reply to this message or contact the journal office at plosone@plos.org. Please include the following items when submitting your revised manuscript:A rebuttal letter that responds to each point raised by the academic editor and reviewer(s). You should upload this letter as a separate file labeled 'Response to Reviewers'.A marked-up copy of your manuscript that highlights changes made to the original version. You should upload this as a separate file labeled 'Revised Manuscript with Track Changes'.An unmarked version of your revised paper without tracked changes. You should upload this as a separate file labeled 'Manuscript'.

We look forward to receiving your revised manuscript.

Kind regards,

Barbara Dritschel, PhD

Academic Editor

PLOS ONE

Reviewers' comments:

Reviewer's Responses to Questions

**Comments to the Author**

1. If the authors have adequately addressed your comments raised in a previous round of review and you feel that this manuscript is now acceptable for publication, you may indicate that here to bypass the “Comments to the Author” section, enter your conflict of interest statement in the “Confidential to Editor” section, and submit your "Accept" recommendation.

Reviewer #1: All comments have been addressed

Reviewer #2: (No Response)

Reviewer #3: (No Response)

2. Is the manuscript technically sound, and do the data support the conclusions?

Reviewer #1: Yes

Reviewer #2: No

Reviewer #3: Yes

3. Has the statistical analysis been performed appropriately and rigorously? 

Reviewer #1: Yes

Reviewer #2: Yes

Reviewer #3: Yes

4. Have the authors made all data underlying the findings in their manuscript fully available?

Reviewer #1: Yes

Reviewer #2: Yes

Reviewer #3: Yes

5. Is the manuscript presented in an intelligible fashion and written in standard English?

Reviewer #1: Yes

Reviewer #2: Yes

Reviewer #3: Yes

6. Review Comments to the Author

Reviewer #1: (No Response)

Reviewer #2: MS: PONE-D-23-08778R1

TI: Is there a bilingual advantage in auditory attention among children: A systematic review and meta-analysis of standardized tests

Review: The original ms is stronger than the revised ms. Things are moving in the wrong direction. The primary weakness remains the motivation for the study. This and other concerns are detailed below.

1. In their ms, the authors claimed that they want to stay clear of the debate concerning a general bilingual advantage, which is fair. However, with the current revision, the authors place themselves squarely in the middle of the debate and they come across as pretty tone-deaf to what is happening in the field. The last 3 or 4 years have witnessed the publication of several very comprehensive meta-analyses that all conclude there is little evidence that children’s language status is related to executive attention, or executive function more broadly, and that the small amount of evidence that does exist has been helped along by publication bias. Rather than give serious consideration to this evidence, the authors pretend like none of this was ever published. The title of their paper provocatively invokes the notion that bilinguals might be advantaged relative to monolinguals. Then in the first paragraph (lines 46 to 53) of the Intro, they effectively claim that decades of research suggest bilinguals are advantaged in “conflict resolution” and executive control tasks relative to monolinguals. This is simply wrong. The conclusion to be drawn from Gunnerud’s and Lowe’s meta-analyses is that there is NO evidence of any difference between monolingual and bilinguals in attention functions or EF more broadly. NONE. That should be starting point. Claiming that they want to steer clear of the debate but then arguing that bilingualism can influence attention function is disingenuous. This is what the argument has been about – and the supporting evidence is pretty weak. RECOMMENDATION: Look to see whether either Gunnerud or Lowe reviewed studies in which children were administered standardized auditory attention measures, and if not, argue that it might be worth examining. Don't argue you want to stay clear of the debate and then hypothesize that bilinguals will be advantaged in attention. That is not steering clear of the debate.

2. Not sure the argument that attention has not been clearly defined is all that persuasive (lines 75). Bialystok has gone to great lengths distinguishing between selective attention, executive attention, and attentional inertia, and has been very clear that the impact of bilingualism on attention function should be confined to executive but not selective attention. Recent findings from developmental studies and meta-analysis suggest there is no difference between monolinguals and bilinguals within any of these domains of attention function but that is besides the point. Attention as a concept has been quite rigorously defined in the literature on bilingualism. The statement on line 75 is therefore dubious. RECOMMENDATION: Conduct a proper review of the definitions of attention that have been put forward by bilingualism researchers (e.g., Bialystok). They are not that vague.

3. It is remarkable that in discussing the fact there were no differences between monolinguals or bilinguals in measures of auditory attention, the authors do not draw any parallels between their findings and those of Gunnerud or Lowe. Not even a reference to these papers. Instead, the first paragraph of the Discussion ends with a vague assertion that their findings may contribute to the development of a theory of the bilingual advantage (line 325 – 328). I’m sorry, but what? How will null findings concerning differences between monolinguals and bilinguals contribute to a theory of the bilingual advantage? Are these findings not more consistent with the conclusions of Gunnerud and Lowe (unreferenced) that there are no differences between monolinguals and bilinguals in measures of attention? RECOMMENDATION: Call a spade a spade. According to the Results, there were no differences. The conclusion should be that there is little evidence for differences.

4. Still not clear why measuring bilingualism as a continuum will reveal effects that are not revealed by the study of extreme groups (line 374 to 385). This question came up in the last review and remains unaddressed in the current revision.

Reviewer #3: As the statistical reviewer I will focus on methods and reporting

Major

1) the quality of the studies has not been assessed using a standard tool for observational studies. this needs to be delivered.

2) the statistical analysis section needs to be restructured, start with the meta analysis, the heterogeneity etc and finish with the meta-regression (but more on all these, specifically in other points).

3) Meta-regression is a stab in the dark usually and is underpowered to detect anything but massive associations (effectively a regression with X observations, where X is the number of available studies). You should discuss this as a major limitation. Even with 60 or 80 studies, it can provide little insight.

4) clearly report how you quantified heterogeneity in the methods section. Also report the confidence intervals for I^2 as argued in http://www.ncbi.nlm.nih.gov/pubmed/17974687. A simple formula exists in the seminal 2002 Higgins paper that proposed I^2.

Minor

1) abstract: add some information on methods, random-effects model? how heterogeneity was quantified? was publication bias assessed and how? how was quality of the studies assessed?

2) how were bilingual families defined? one parent bilingual? both?

3) Year may be worth considering in bias assessmen: http://www.ncbi.nlm.nih.gov/pubmed/25988604. With newer studies we would be more confident.

4) clarify that a random effects model was used in the methods section? was it a DerSimonial-Laird RE model with inverse variance weighting? please clarify.

7. PLOS authors have the option to publish the peer review history of their article (what does this mean?). If published, this will include your full peer review and any attached files.

Reviewer #1: No

Reviewer #2: No

Reviewer #3: No

---

## [Author Response · Author response to Decision Letter 1]

28 Nov 2023

Since no comments were provided by the original first reviewer, we followed the Editor’s reference to the other two reviewers as Reviewer #1 and Reviewer #2, respectively. We thank both of them for their valuable feedback and thoughtful suggestions. Please see our point-to-point response to each comment below.

Reviewer #1

The original ms is stronger than the revised ms. Things are moving in the wrong direction. The primary weakness remains the motivation for the study. This and other concerns are detailed below.

Comment 1.1:

In their ms, the authors claimed that they want to stay clear of the debate concerning a general bilingual advantage, which is fair. However, with the current revision, the authors place themselves squarely in the middle of the debate and they come across as pretty tone-deaf to what is happening in the field. The last 3 or 4 years have witnessed the publication of several very comprehensive meta-analyses that all conclude there is little evidence that children’s language status is related to executive attention, or executive function more broadly, and that the small amount of evidence that does exist has been helped along by publication bias. Rather than give serious consideration to this evidence, the authors pretend like none of this was ever published. The title of their paper provocatively invokes the notion that bilinguals might be advantaged relative to monolinguals. Then in the first paragraph (lines 46 to 53) of the Intro, they effectively claim that decades of research suggest bilinguals are advantaged in “conflict resolution” and executive control tasks relative to monolinguals. This is simply wrong. The conclusion to be drawn from Gunnerud’s and Lowe’s meta-analyses is that there is NO evidence of any difference between monolingual and bilinguals in attention functions or EF more broadly. NONE. That should be starting point. Claiming that they want to steer clear of the debate but then arguing that bilingualism can influence attention function is disingenuous. This is what the argument has been about – and the supporting evidence is pretty weak. RECOMMENDATION: Look to see whether either Gunnerud or Lowe reviewed studies in which children were administered standardized auditory attention measures, and if not, argue that it might be worth examining. Don't argue you want to stay clear of the debate and then hypothesize that bilinguals will be advantaged in attention. That is not steering clear of the debate.

Response 1.1: 

Thank you for the feedback. As opposed to the Reviewer’s observation, we were fully aware of the meta-analyses published in the past few years (e.g., Donnelly et al., 2019; Lehtonen, 2018), but please note that most of these meta-analyses are with adults whereas the focus of our manuscript is the developmental population. In the manuscript, we cited two more recent and comprehensive ones that focused on children—Gunnerud et al. (2020) and Lowe et al. (2021)—as examples. We appreciate the Reviewer’s suggestion and have added that “It is worth noting that when addressing attention, nearly all studies included in these meta-analyses have focused on the (audio-)visual domain, and very few measure auditory attention through tools like behavioral tasks or standardized tests.” Meanwhile, we believe that it is crucial to present a full picture, especially when addressing such a controversial topic, to the wide readership of PLOS ONE. Since not every reader is familiar with the bilingual advantage hypothesis, we explained this concept in the first paragraph, and provided two examples about conflict resolution and executive control. Note that we also cited Dunabeitia et al. (2014) to illustrate equal performance (in inhibitory tasks) between monolingual and bilingual children. 

Furthermore, we referenced Grundy (2020) who used the Bayesian statistical approach and suggested that “when group differences do appear on EF tasks, bilinguals outperform monolinguals far more likely than chance.” Importantly, Grundy (2020, p. 177) pointed out that these findings “highlight the need to determine when, rather than if, bilinguals outperform monolinguals on EF tasks”, which echoes with our statement that it remains unclear when a bilingual advantage occurs, as delineated at the end of the second paragraph in the manuscript.

In addition, we would like to emphasize that we exercised extreme care to remain impartial in describing hypotheses and making claims throughout the manuscript. For example, it is based on empirical evidence on bilingualism modulating audiovisual speech processing (Pons et al., 2015; not on the bilingual advantage per se) that we predicted “if auditory attention development is shaped by bilingual experience, bilingual children might have more accurate and/or faster responses than their monolingual counterparts in standardized tests.” Critically, keeping clear of a dichotomous debate of the bilingual advantage does not mean that we should and we can ignore the bilingualism effects (put it simply, the bilingual advantage ≠ the bilingualism effects). As we already stated in the third paragraph of the paper, “we acknowledge that bilingualism can exert influence on cognition at least in some types of bilinguals, which has been supported by empirical studies …” It would be appreciated if the Reviewer could be more explicit regarding how we could have formed our hypothesis better. 

References:

Donnelly, S., Brooks, P. J., & Homer, B. D. (2019). Is there a bilingual advantage on interference-control tasks? A multiverse meta-analysis of global reaction time and interference cost. Psychonomic Bulletin and Review, 26(4), 1122–1147.

Lehtonen, M., Soveri, A., Laine, A., Järvenpää, J., De Bruin, A., & Antfolk, J. (2018). Is bilingualism associated with enhanced executive functioning in adults? A meta-analytic review. Psychological Bulletin, 144(4), 394–425.

Comment 1.2: 

Not sure the argument that attention has not been clearly defined is all that persuasive (lines 75). Bialystok has gone to great lengths distinguishing between selective attention, executive attention, and attentional inertia, and has been very clear that the impact of bilingualism on attention function should be confined to executive but not selective attention. Recent findings from developmental studies and meta-analysis suggest there is no difference between monolinguals and bilinguals within any of these domains of attention function but that is besides the point. Attention as a concept has been quite rigorously defined in the literature on bilingualism. The statement on line 75 is therefore dubious. RECOMMENDATION: Conduct a proper review of the definitions of attention that have been put forward by bilingualism researchers (e.g., Bialystok). They are not that vague.

Response 1.2: 

Thank you for the recommendation. We had properly reviewed definitions of attention prior to drafting the manuscript, as it has been an important component of our work. Our statement that attention is vaguely defined in the cognitive literature is supported by our search (see Bialystok & Craik, 2022; Styles, 2006).

We had also conducted a thorough search on how attention has been defined by bilingualism researchers, especially by Bialystok. It is clear that how attention was conceptualized in bilingualism literature has been heavily shaped by attention models in psychology (e.g., Enns, 1990; Posner & Petersen, 1990). It is also important to note the difference between bilingualism researchers using/discussing different attention related concepts (e.g., selective attention, executive attention) and clearly defining them. For example, when discussing “selective attention”, Bialystok (1992) related it to selectivity in Enns’s (1990) visual attention scheme, where it represents the highest form of attention that deals with the limitations of the sensory systems and the finite capacity of brain processes, but she did not provide a clear definition about selective attention. Another essential term is “executive attention”. In Bialystok (2017, p. 250), she mentioned that “the notion of executive attention incorporates elements from executive function models and from attention accounts.” Specifically, she used working memory capacity (i.e., a combination of working memory and attention, proposed by Engle and colleagues; see Shipstead et al., 2015) as an equal construct to executive attention. However, she did not define executive attention directly either. 

Rather, attention has not been more clearly conceptualized until in a recent article, Bialystok and Craik (2022), which adopted another concept—“attentional control”—to explain the mechanism behind how bilingualism impacts cognition. Attentional control was broadly described as “a repertoire of processing operations that specific tasks and higher-level cognitive functions can utilize to fulfill their various goals” (Bialystok & Craik, 2022, p. 1253), and composed of procedures that either facilitate mental operations (e.g., selection, goal maintenance, coordination, engagement, disengagement) or inhibit mental operations (e.g., interference suppression, response inhibition). Further, Bialystok and Craik (2022, p. 1260) suggested that “lifelong bilingualism has conferred enhanced levels of attentional control to speakers of two or more languages, providing a robust basis for a range of cognitive tasks, including those dependent on executive functions.” 

Therefore, prior to Bialystok and Craik (2022), attention as a concept was not rigorously defined in bilingualism research. To make it clearer, we have added in the manuscript that “In bilingualism research, attention was not clearly conceptualized prior to a recent paper by Bialystok and Craik [16]. According to it, lifelong bilingual experience enhances attentional control, which is defined as a repertoire of processing operations that higher-level cognition utilizes to fulfill various goals [16].” Our observation about the bilingualism literature is that researchers tend to use various executive functioning tasks to examine the bilingualism effects, some of which are related to attention, such as the Dimensional Change Card Sort task (e.g., Bialystok & Martin, 2004) and the flanker task (e.g., Sorge et al., 2017). However, it remains a question even to what extent these tasks measure attention along with its components (rather than other overlapping constructs like working memory, etc.) and what aspects of attention they measure exactly. Yet it has been a common practice that researchers interpret their results around certain attentional concepts (e.g., selective attention, executive attention, attention control, etc.) and integrate with other empirical findings. In part, this could have contributed to the accumulating contradictory findings in the field, which sometimes result from inconsistent definitions of a given construct. 

That being said, we fully acknowledge the foundational and inspiring work of Bialystok and colleagues in exploring the association between bilingualism and attention, and it is attentional control that lifelong bilingualism impacts (Bialystok & Craik, 2022). Though in our manuscript, we used the attention accounts developed by Posner and colleagues because they have a relatively clear categorization of attention components that are also found to be represented in different brain regions. Further, this account might be more appropriate when we consider components of auditory attention, which is the focus of our paper. 

References:

Bialystok, E. (1992). Selective attention in cognitive processing: The bilingual edge. In R. J. Harris (Ed.), Cognitive processing in bilinguals (pp. 501-513). North Holland: Elsevier Science Publishers. 

Bialystok, E., & Martin, M. M. (2004). Attention and inhibition in bilingual children: evidence from the dimensional change card sort task. Developmental Science, 7(3), 325-339.

Bialystok, E., & Craik, F. I. M. (2022). How does bilingualism modify cognitive function? Attention to the mechanism. Psychonomic Bulletin & Review, 29, 1246-1269.

Enns, J. T. (1990). Relations between components of visual attention. In J. T. Enns (Ed.), The development of attention: Research and theory (pp. 139-158). North Holland: Elsevier Science Publishers. 

Shipstead, Z., Harrison, T. L., & Engle, R. W. (2015). Working memory capacity and the scope and control of attention. Attention, Perception, & Psychophysics, 77, 1863-1880.

Sorge, G. B., Toplak, M. E., & Bialystok, E. (2017). Interactions between levels of attention ability and levels of bilingualism in children’s executive functioning. Developmental Science, 20, e12408. 

Styles, E. A. (2006). The psychology of attention. Psychology Press.

Comment 1.3: 

It is remarkable that in discussing the fact there were no differences between monolinguals or bilinguals in measures of auditory attention, the authors do not draw any parallels between their findings and those of Gunnerud or Lowe. Not even a reference to these papers. Instead, the first paragraph of the Discussion ends with a vague assertion that their findings may contribute to the development of a theory of the bilingual advantage (line 325 – 328). I’m sorry, but what? How will null findings concerning differences between monolinguals and bilinguals contribute to a theory of the bilingual advantage? Are these findings not more consistent with the conclusions of Gunnerud and Lowe (unreferenced) that there are no differences between monolinguals and bilinguals in measures of attention? RECOMMENDATION: Call a spade a spade. According to the Results, there were no differences. The conclusion should be that there is little evidence for differences.

Response 1.3: 

Thank you very much for the helpful recommendation. In order to improve clarity and draw a parallel, we have explicitly stated our conclusion based on the results and connected them to the findings of Gunnerud et al. (2020) and Lowe et al. (2021). Specifically, we added the following text in Discussion: “Overall, there was little difference between monolingual and bilingual children’s performance on standardized auditory attention tests. This finding is consistent with those of Gunnerud et al. [7] and Lowe et al. [8], though both of them analyzed tasks focusing on visual attention.”

In addition, we have revised the previous sentence about theory development as “Our work, however, contributes to the field by uncovering the relation between bilingualism and auditory attention.” We hope this revision reflects the significance of our work more accurately. 

Comment 1.4: 

Still not clear why measuring bilingualism as a continuum will reveal effects that are not revealed by the study of extreme groups (line 374 to 385). This question came up in the last review and remains unaddressed in the current revision.

Response 1.4: 

Thank you for the feedback. In Discussion, we emphasized the importance of assessing bilingualism as a continuum in multiple places, because only by doing so, can we capture the nuance within the bilingual populations and provide a fuller picture of the bilingualism effects. As we already explained in the manuscript, evidence suggests that bilingual variables such as language proficiency, age of acquisition can affect cognitive performance (Gollan et al., 2005; Festman et al., 2022; Tse & Altarriba, 2014). “For example, more proficient bilinguals tend to exhibit a positive influence of bilingualism. Consistent with this, we observed a trend of the bilingual advantage toward adolescents relative to younger children, … ” In other words, a dichotomous view of participants as monolingual vs. bilingual overlooks the individual difference, particularly within bilinguals. The story that bilinguals can tell us is probably way richer than that unfolded by simply comparing them with their monolingual counterparts. 

Reviewer #2

As the statistical reviewer I will focus on methods and reporting

Response: We greatly appreciate your feedback on our methods and statistical reporting, which has significantly improved the quality of our manuscript.

Major

Comment 2.1:

1) the quality of the studies has not been assessed using a standard tool for observational studies. this needs to be delivered.

Response 2.1:

---

## [Decision Letter · Decision Letter 2]

9 Jan 2024

PONE-D-23-08778R2Is there a bilingual advantage in auditory attention among children? A systematic review and meta-analysis of standardized auditory attention testsPLOS ONE

Dear Dr. Bao,

Thank you for submitting your manuscript to PLOS ONE. After careful consideration, we feel that it has merit but does not fully meet PLOS ONE’s publication criteria as it currently stands. Therefore, we invite you to submit a revised version of the manuscript that addresses the points raised during the review process. All reviewers and myself have read your  revised manuscript and feel that it is  much improved. Reviewer 2 has pointed out that there are still a few statements that are either inaccurate biased  or confusing.  They are listed below. You should review these statements and revise them or present a good argument for why there reviewer is incorrect.   

We look forward to receiving your revised manuscript.

Kind regards,

Barbara Dritschel, PhD

Academic Editor

PLOS ONE

Journal Requirements:

Reviewers' comments:

Reviewer's Responses to Questions

**Comments to the Author**

1. If the authors have adequately addressed your comments raised in a previous round of review and you feel that this manuscript is now acceptable for publication, you may indicate that here to bypass the “Comments to the Author” section, enter your conflict of interest statement in the “Confidential to Editor” section, and submit your "Accept" recommendation.

Reviewer #2: (No Response)

Reviewer #3: All comments have been addressed

2. Is the manuscript technically sound, and do the data support the conclusions?

Reviewer #2: Partly

Reviewer #3: Yes

3. Has the statistical analysis been performed appropriately and rigorously? 

Reviewer #2: Yes

Reviewer #3: Yes

4. Have the authors made all data underlying the findings in their manuscript fully available?

Reviewer #2: Yes

Reviewer #3: Yes

5. Is the manuscript presented in an intelligible fashion and written in standard English?

Reviewer #2: Yes

Reviewer #3: Yes

6. Review Comments to the Author

Reviewer #2: The paper is much improved. However, there are a few remaining statements that are either inaccurate, highly biased, or confusing. These should be revised.

1. In the Abstract, the authors refer to the “effects” of bilingualism on cognition (line 25). This is misleading. All research in this field is based on cross-sectional between-subjects comparisons, so at best, current knowledge concerns associations between bilingualism and cognition.

2. Line 26. “Developmental studies reveal different cognitive profiles between monolinguals and bilinguals in (audio)-visual attention tasks.” This is a biased and misleading characterization of the literature. Current meta-analyses clearly indicate there are no differences between monolingual and bilingual children in attention tasks. Also, what is an “(audio)-visual” attention task? Why not “audio-visual”?

3. Line 367. “Our work…contributes to the field by uncovering the relation between bilingualism and auditory attention.” How does this statement hang together with the sentence on line 360 stating that the bilingual advantage likely does not exist? Based on the results they are reporting, what the authors have uncovered is that there is no relation. Confusing.

4. Line 366. The authors write that there is no theoretical framework in which they can place their findings. Ken Paap just wrote an entire book on the bilingual advantage hypothesis and the replication crisis in psychology. Null findings like the ones reported in this analysis fit very well within the framework discussed in his book.

https://www.routledge.com/The-Bilingual-Advantage-in-Executive-Functioning-Hypothesis-How-the-debate/Paap/p/book/9781032310992

Reviewer #3: I am satisfied with the authors' responses and the resulting changes to the paper, which, in my opinion, reads much better now.

7. PLOS authors have the option to publish the peer review history of their article (what does this mean?). If published, this will include your full peer review and any attached files.

Reviewer #2: No

Reviewer #3: No

---

## [Author Response · Author response to Decision Letter 2]

15 Jan 2024

Once again, we would like to thank both reviewers for their valuable feedback, which has significantly improved our paper. In particular, we have addressed the four comments of Reviewer #2 and revised the manuscript accordingly. Please see our point-to-point response below.

Reviewer #2

The paper is much improved. However, there are a few remaining statements that are either inaccurate, highly biased, or confusing. These should be revised.

1. In the Abstract, the authors refer to the “effects” of bilingualism on cognition (line 25). This is misleading. All research in this field is based on cross-sectional between-subjects comparisons, so at best, current knowledge concerns associations between bilingualism and cognition.

Response: Thank you for the feedback. Following the Reviewer’s suggestion, we have revised it as “A wealth of research has investigated the associations between bilingualism and cognition, especially in regards to executive function.” 

Nevertheless, it should be noted that many researchers have referred to the “effects” of bilingualism on cognition, which is widely acceptable in the field. Over decades, for example, Bialystok and colleagues have consistently referred to it in myriads of their publications (e.g., Anderson et al., 2018; Bialystok, 1997, 2007, 2021). Likewise, Paap (2023) has referred to it throughout his book, which was suggested by the Reviewer in another comment. 

We would also would like to note that not “all research in this field is based on cross-sectional between-subjects comparisons.” Such a view overlooks the great number (although smaller than cross-sectional designs) and value of longitudinal studies in bilingualism research. For instance, in one of the studies we reviewed in the manuscript, Nicoley and Poncelet (2015) followed monolingual and bilingual kindergarten children for three years and investigated the longitudinal effect of bilingual immersion schooling on cognition (also see Woumans et al., 2016). An example of adult study is Sörman et al. (2017), who tested monolingual and bilingual older adults on dual-tasking over six waves from 1988 to 2014. 

References:

Anderson, J. A. E., Grundy, J. G., De Frutos, J., Barker, R. M., Grady, C., & Bialystok, E. (2018). Effects of bilingualism on white matter integrity in older adults. NeuroImage, 167, 143–150. https://doi.org/10.1016/j.neuroimage.2017.11.038

Bialystok, E. (1997). Effects of bilingualism and biliteracy on children’s emerging concepts of print. Developmental Psychology, 33(3), 429–440. https://doi.org/10.1037/0012-1649.33.3.429

Bialystok, E. (2007). Cognitive effects of bilingualism: How linguistic experience leads to cognitive change. International Journal of Bilingual Education and Bilingualism, 10(3), 210–223. https://doi.org/10.2167/beb441.0

Bialystok, E. (2021). Cognitive effects of bilingualism: An evolving perspective. In W. S. Francis (Ed.), Bilingualism across the lifespan: Opportunities and challenges for cognitive research in a global society (1st ed., pp. 9–28). New York: Routledge.

Paap, K. (2023). The bilingual advantage in executive functioning hypothesis: How the debate provides insight into psychology’s replication crisis. New York: Routledge.

Sörman, D. E., Josefsson, M., Marsh, J. E., Hansson, P., & Ljungberg, J. K. (2017). Longitudinal effects of bilingualism on dual-tasking. PLOS ONE, 12(12), e0189299. https://doi.org/10.1371/journal.pone.0189299

Woumans, E., Surmont, J., Struys, E., & Duyck, W. (2016). The longitudinal effect of bilingual immersion schooling on cognitive control and intelligence. Language Learning, 66(S2), 76–91. https://doi.org/10.1111/lang.12171

2. Line 26. “Developmental studies reveal different cognitive profiles between monolinguals and bilinguals in (audio)-visual attention tasks.” This is a biased and misleading characterization of the literature. Current meta-analyses clearly indicate there are no differences between monolingual and bilingual children in attention tasks. Also, what is an “(audio)-visual” attention task? Why not “audio-visual”?

Response: Thank you for the feedback. In the Introduction, we reviewed developmental studies that used methods other than behavioral attention tasks, which are the focus of “current meta-analyses,” and suggested different cognitive outcomes between monolinguals and bilinguals. For example, using event-related potentials, Kuipers and Thierry (2015) found that bilingual toddlers display greater attention to speech than monolingual peers. Similarly, using EEG, Nacar Garcia et al. (2018) found that bilingual infants show increased attention to the speech signal than monolinguals. To reflect this, we have specified in the revised sentence that “some” studies revealed differences between monolinguals and bilinguals.

By “(audio)-visual”, we meant visual or audio-visual. To improve clarity, we have revised the relevant sentence as “Some developmental studies reveal different cognitive profiles between monolinguals and bilinguals in visual or audio-visual attention tasks …”

Reference:

Nacar Garcia, L., Guerrero-Mosquera, C., Colomer, M., & Sebastian-Galles, N. (2018). Evoked and oscillatory EEG activity differentiates language discrimination in young monolingual and bilingual infants. Scientific Reports, 8, 2770. https://doi.org/10.1038/s41598-018-20824-0

3. Line 367. “Our work…contributes to the field by uncovering the relation between bilingualism and auditory attention.” How does this statement hang together with the sentence on line 360 stating that the bilingual advantage likely does not exist? Based on the results they are reporting, what the authors have uncovered is that there is no relation. Confusing.

Response: Thank you for the feedback. To improve clarity, we have revised the sentence as “Our work specifically suggests that a bilingual advantage is not observable when measured through standardized auditory attention tests.”

4. Line 366. The authors write that there is no theoretical framework in which they can place their findings. Ken Paap just wrote an entire book on the bilingual advantage hypothesis and the replication crisis in psychology. Null findings like the ones reported in this analysis fit very well within the framework discussed in his book. https://www.routledge.com/The-Bilingual-Advantage-in-Executive-Functioning-Hypothesis-How-the-debate/Paap/p/book/9781032310992

Response: Thank you very much for the suggestion. We have referenced Paap’s book and revised the relevant sentence as “Importantly, our result fits well within the theoretical framework discussed in Paap [59], who highlights that the bilingual advantage either does not exist or is restricted to very specific circumstances.”

---

## [Editor Report · Decision Letter 3]

9 Feb 2024

Is there a bilingual advantage in auditory attention among children? A systematic review and meta-analysis of standardized auditory attention tests

PONE-D-23-08778R3

Dear Dr. Bao,

We’re pleased to inform you that your manuscript has been judged scientifically suitable for publication and will be formally accepted for publication once it meets all outstanding technical requirements.

Kind regards,

Barbara Dritschel, PhD

Academic Editor

PLOS ONE